# An Experimental Field Trial Investigating the Use of Bacteriophage and Manure Slurry Applications in Beef Cattle Feedlot Pens for *Salmonella* Mitigation

**DOI:** 10.3390/ani13203170

**Published:** 2023-10-11

**Authors:** Colette A. Nickodem, Ashley N. Arnold, Matthew R. Beck, K. Jack Bush, Kerri B. Gehring, Jason J. Gill, Tram Le, Jarret A. Proctor, John T. Richeson, H. Morgan Scott, Jason K. Smith, T. Matthew Taylor, Javier Vinasco, Keri N. Norman

**Affiliations:** 1Department of Veterinary Integrative Biosciences, Texas A&M University, College Station, TX 77843, USA; cnickodem@wisc.edu; 2Texas A&M Veterinary Diagnostic Laboratory, Texas A&M University, College Station, TX 77843, USA; a.arnold@tamu.edu; 3United States Department of Agriculture-Agriculture Research Service, Bushland, TX 79012, USA; matthew.r.beck@usda.gov; 4Texas A&M AgriLife Research, Texas A&M University System, Amarillo, TX 79106, USA; jack.bush@ag.tamu.edu; 5Department of Animal Science, Texas A&M University, College Station, TX 77843, USA; kbgehring@tamu.edu (K.B.G.); jason.gill@ag.tamu.edu (J.J.G.); tram.le@ag.tamu.edu (T.L.); matt.taylor@ag.tamu.edu (T.M.T.); 6Center for Phage Technology, Texas A&M University, College Station, TX 77845, USA; 7Department of Agricultural Sciences, West Texas A&M University, Canyon, TX 79016, USA; jricheson@wtamu.edu; 8Department of Veterinary Pathobiology, Texas A&M University, College Station, TX 77843, USA; hmscott@cvm.tamu.edu (H.M.S.); jvinasco-torres@cvm.tamu.edu (J.V.); 9Texas A&M AgriLife Extension, Department of Animal Science, Texas A&M University, Amarillo, TX 79106, USA; jason.smith@ag.tamu.edu

**Keywords:** pre-harvest, *Salmonella*, cattle, feedlots, food safety, bacteriophage

## Abstract

**Simple Summary:**

Most post-harvest interventions for addressing *Salmonella* in beef products are applied directly to the carcass surface and are ineffective at reducing or eliminating *Salmonella* that are harbored in cattle lymph nodes. Bacteriophages are viruses that are ubiquitous in the environment, including agricultural facilities, that only infect and destroy bacterial cells. The objectives of this experiment were to determine whether natural bacteriophage populations contained in manure slurry or laboratory-curated bacteriophage cocktails could reduce *Salmonella* prevalence on feedlot pen surfaces and on cattle hides and subsequently in cattle lymph nodes. Applications of manure slurry and bacteriophage cocktail treatments reduced the prevalence of *Salmonella* on cattle hides which may contribute to lymph node contamination. Lymph nodes may become incorporated into ground beef products, which can contribute to beef-related foodborne illness, and therefore, these treatments have the potential to address food safety concerns.

**Abstract:**

Post-harvest *Salmonella* mitigation techniques are insufficient at addressing *Salmonella* harbored in cattle lymph nodes, necessitating the exploration of pre-harvest alternatives that reduce *Salmonella* prior to dissemination to the lymph nodes. A 2 × 2, unbalanced experiment was conducted to determine the effectiveness of pre-harvest treatments applied to the pen surface for *Salmonella* mitigation in cattle. Treatments included manure slurry intended to mimic pen run-off water (n = 4 pens), a bacteriophage cocktail (n = 4), a combination of both treatments (n = 5), and a control group (n = 5) that received no treatment. Environment samples from 18 feedlot pens and fecal grabs, hide swabs, and subiliac lymph nodes from 178 cattle were collected and selectively enriched for *Salmonella*, and *Salmonella* isolates were sequenced. The combination treatment was most effective at reducing *Salmonella*, and the prevalence was significantly lower compared with the control group for rump swabs on Days 14 and 21. The treatment impact on *Salmonella* in the lymph nodes could not be determined due to low prevalence. The reduction on cattle hides suggests that bacteriophage or water treatments applied to the feedlot pen surface may reduce *Salmonella* populations in cattle during the pre-harvest period, resulting in reduced contamination during slaughter and processing.

## 1. Introduction

The United States has the largest feedlot cattle industry in the world [1]. This industry supports U.S. consumers and beef exports, which were at a record 3.5 billion pounds high in 2022 [1]. Recently, efforts have focused on using sustainable methods that ensure safe product availability in the beef market [2]. Cattle in the U.S. are typically raised at calf–cow ranches, transported to backgrounding facilities to put on weight, and finally sent to feedlots for grain finishing [2]. Good animal welfare practices can improve animal health, which is associated with providing safe, pathogen-free beef products [3]. Interventions in this pre-harvest period can help mitigate *Salmonella* prior to harvest and increase food safety.

The prevalence of foodborne illnesses from *Salmonella*-contaminated beef products has shifted over time from roasts or whole cuts to ground beef [4]. This prevalence reduction in whole-muscle beef cuts is likely attributed to the availability and widespread use of *Salmonella*-reducing post-harvest interventions applied to cattle carcasses [5,6,7]. These practices include but are not limited to evisceration, antimicrobial sprays, and thermal treatments for carcass and surface pathogen mitigation [8,9,10,11,12]. *Salmonella* in ground beef is thought to originate from *Salmonella* harbored in cattle lymph nodes that are difficult to separate from adipose tissues and evade external carcass pathogen mitigation techniques [13]. In 2022, the USDA Food Safety Inspection Service (FSIS) identified *Salmonella* in 2.1% (144/11,009) of raw ground beef samples collected from establishments (n = 1231) in their verification testing program [14]. There has been little progress in reducing *Salmonella* in ground beef; however, pre-harvest alternatives have the potential to address *Salmonella*, prior to dissemination into the lymph nodes. Examples of pre-harvest options that may be administered directly to the animal include direct-fed microbials, vaccinations, or bacteriophages, all with varying effectiveness [15,16,17]. *Salmonella* carriage in cattle is often related to the *Salmonella* populations in the feedlot environment [18,19]. Distinct *Salmonella* populations have been observed in cattle originating from different feedlot pens, indicating that *Salmonella* control within the environment may also impact *Salmonella* populations within cattle [20].

Bacteriophages (phages) are bacterial viruses that must utilize bacterial host machinery to replicate and produce new progeny. Phages are safe to use as antibiotic alternatives in food production or therapeutically for humans and animals with limited adverse events because of their host specificity to bacteria [21]. Bacteriophages and bacteriophage cocktails have several advantageous qualities for reducing bacteria in agricultural settings: (1) the phage will have minimal effect on the normal flora of the livestock, (2) a specific phage can be selected to target pathogenic bacteria commonly identified in cattle or the feedlot environment, (3) the use of phage cocktails can increase the host range to address multiple serovars of public health concern, (4) and the use of phages may counteract antimicrobial resistance (AMR) by increasing bacterial sensitivity to antibiotics [21,22,23,24,25].

Post-harvest bacteriophage use has been successful at reducing *Salmonella* in food products including milk, chicken, and beef [26,27,28,29,30]. Alternatively, pre-harvest bacteriophage applications in the swine and poultry industries administered by a variety of delivery methods (i.e., intranasal, oral gavage, or aerosol spray) have shown *Salmonella* reductions in animal cecum contents [31,32,33]. There is limited pre-harvest bacteriophage research addressing *Salmonella* in beef and dairy cattle; however, the use of phages for reducing other pathogens such as *Escherichia coli* O157:H7 and *Staphylococcus aureus* has been investigated [34,35]. A recent study demonstrated the arrival of injected phages to cattle peripheral lymph nodes but there was not a significant reduction in *Salmonella,* suggesting that phage–bacteria interactions at this site of infection may not be optimal [17]. However, naturally occurring bacteriophages have been successful at reducing Enterobacteriaceae that were identified in the environment and animal samples at swine and poultry farms [36,37]. *Salmonella* phages were also found to be common in the environment at Texas feedlots and have been isolated from manure samples from dairy farms in New York [38,39]. These studies provide evidence that naturally occurring phages are present in the agricultural environment and have the potential to be harnessed for *Salmonella* mitigation; however, this concept has not been applied in the feedlot setting.

The objective of this experiment was to test two treatments applied in the feedlot environment, including manure slurry (which was presumed to contain environmental phages) applied through a dust abatement sprinkler system and a targeted bacteriophage cocktail applied using battery-operated backpack sprayers. The purpose of the manure slurry treatment was to harness and enhance the presence of bacteriophages naturally present in the feedlot environment. The bacteriophage cocktail utilized laboratory-curated stocks of phages known to infect *Salmonella* serovars previously identified in a research feedlot. We hypothesized that both treatments would reduce the presence of *Salmonella* in the feedlot environment, and due to normal cattle movement and grooming behavior, there would be a subsequent reduction in *Salmonella* present on cattle hide, in feces, and in subiliac lymph nodes.

## 2. Materials and Methods

### 2.1. Study Design

The present experiment was conducted to test the effect of two treatments applied to feedlot pen surfaces to reduce *Salmonella* in the feedlot environment, on beef cattle hide, within feces, and in subiliac lymph nodes. All procedures involving animals were approved by and conducted in accordance with guidelines of the West Texas A&M University Institutional Animal Care and Use Committee. Treatments were applied to the pen surface in a 2 × 2 full factorial arrangement, with an unbalanced design which included manure slurry that was applied with a dust abatement sprinkler system and a targeted bacteriophage cocktail that was applied with backpack sprayers. The study had three parts: a 36-day pre-trial that included 3 full sample collections at the West Texas A&M Research Feedlot (WTAMU Feedlot), a 21-day experimental trial that included 4 full sample collections at the joint Texas A&M AgriLife/USDA-ARS Research Feedlot (AgriLife Feedlot), and monthly follow-up samplings which occurred from September 2021 to February 2022 at both feedlots (Figure 1). The cattle feedlot locations and details of the specific study parts can be observed in Appendix A. The selection and development of the treatments occurred prior to and during the pre-trial period.

All cattle enrolled in the experiment consumed the same diet which is representative of a typical diet for finishing cattle in the U.S. Southern Great Plains. The feed was manufactured and delivered using the same ingredients and equipment to all treatment groups. All feed ingredients were screened prior to their arrival at the feedlot at their point of origin, according to standard Texas and U.S. guidelines, determined by the Texas Department of Agriculture and the U.S. Food and Drug Administration.

### 2.2. Manure Slurry Treatment

#### 2.2.1. Pen Manure Salmonella Populations

The composite environment manure pack from the pen surface with favorable *Salmonella* (based on serotype and AMR profile) and natural phage populations was selected from the WTAMU Feedlot to create the manure slurry treatment. Whole genome sequencing (WGS) and bacteriophage assays were used to identify the *Salmonella* and phage populations in the composite environment manure pack samples. The samples were selectively enriched for *Salmonella* and *Salmonella* isolates (n = 64), originating from feces and pen environment manure pack samples from the pre-trial sampling period (Days -36 and -21), were sequenced to determine *Salmonella* serovars and to identify antimicrobial resistance (AMR) genes using previously described methods [18]. Specifically, all available environmental *Salmonella* isolates (n = 19) and up to 3 fecal *Salmonella* isolates per pen (n = 45) were selected. Feedlot pens with high *Salmonella* prevalence in the pen environment manure pack or cattle fecal samples, pens with AMR *Salmonella*, or pens with *Salmonella* serovars commonly identified in cattle lymph nodes or of public health concern were excluded from consideration. Ultimately, 4 pens at the WTAMU Feedlot best fit these criteria and were selected for developing the manure slurry treatment.

#### 2.2.2. Pen Manure Phage Presence

Pen composite manure pack samples collected during the pre-trial period from the surface of the 4 WTAMU Feedlot pens were retrospectively tested for the presence of phages. To extract the soil virome, 2 g of each pen environment manure pack sample was added to 10 mL of tryptic soy broth (TSB; Becton, Dickinson and Company (BD), Franklin Lakes, NJ, USA), thoroughly vortexed, and centrifuged for 20 min at 4000 rpm (3399× *g*). The supernatant was filter sterilized through 0.22 µm Fast Flow & Low Binding Millipore Express^®^ PES membrane filters (Millipore Sigma, Burlington, MA, USA). The soil virome extracts were enriched using two separate cocktails composed of 6 *Salmonella* isolates each to select for *Salmonella*-infecting phages and to cover the range of *Salmonella* serovars identified at both feedlots (WTAMU and AgriLife) (Appendix A). Overnight cultures of the *Salmonella* cocktail were sub-cultured in 3 mL of TSB for approximately 3 h with shaking at 37 °C. Next, 2 mL of the soil virome extract was added to the *Salmonella* sub-culture to be enriched during overnight incubation at 37 °C. This was followed by adding 100 µL of chloroform (Macron Fine Chemicals by Avantor, Radnor, PA, USA) to each sample, which were centrifuged at 4000 rpm (3399× *g*) for 5 min, and the supernatant was filter sterilized into new tubes. This enrichment process was repeated three times with both *Salmonella* cocktails for two separate soil virome extract samples from each pen.

To determine the presence and host range of the phages, selectively enriched soil virome samples were plated onto individual *Salmonella* bacterial lawns using spot test methods (Appendix A; isolated from the trial). To accomplish this, 100 µL of each *Salmonella* serovar was incubated overnight at 37 °C in 3 mL of TSB, and 100 µL was transferred to 4 mL of 0.5% top agar (5 g tryptone, 5 g NaCl, 500 mL dH_2_0, and 2.5 g bacto agar). The culture was supplemented with 5 mM MgS0_4_ (Amresco via VWR, Cleveland, OH, USA) and 5 mM CaCl_2_ (Millipore Sigma, Burlington, MA, USA) and poured as an overlay over tryptic soy agar plates (TSA; BD, Franklin Lakes, NJ, USA), followed by spotting 10 µL of each enriched soil virome sample to the lawns and overnight incubation at 37 °C. The plates were observed for clearing zones, signifying the presence of phages.

#### 2.2.3. Manure Slurry Treatment Development and Application

On Day -7 of the pre-trial period, a 20-gallon plastic storage bin (Rubbermaid, Atlanta, GA, USA) of the composite environment manure pack from the surface of each selected pen was collected, shipped to the AgriLife Feedlot, and kept at 4 °C until use. Each Monday afternoon during the experimental trial, manure slurry was prepared by placing approximately one-third of the composite environment manure pack from the storage bin into a double burlap bag and dipping it into a 5000-gallon (18,927.06-L) tank containing well water and allowing it to soak for 5–10 min. The burlap bags allowed small particles to dissolve into the water while retaining larger material that could potentially clog the sprinkler system. This process was repeated for the composite environment manure pack from all 4 pens until the manure slurry was completed. The remaining pen composite environment manure pack samples were kept at 4 °C for use during the following weeks.

The dust abatement sprinkler system at the AgriLife Feedlot is limited to running in blocks of 3 pens at a time; therefore, randomly distributing the manure slurry treatment was not feasible. The 9 easternmost pens were purposefully selected to receive the manure slurry treatment to reduce overspray into non-treatment pens due to common wind patterns. The cattle (n = 10 head per pen) remained in pens during the manure slurry application, which lasted for a 7-min period per block, approximating to one-eighth of an inch (0.32 cm) depth coverage on each pen surface. Upon completion of block 1 (pens 1, 2, and 3), application was repeated sequentially on block 2 (pens 4, 5, and 6) and block 3 (pens 7, 8, and 9). The sprinklers ran daily from Tuesday to Friday mornings each week, followed by a rest period during the weekends. Each application day used approximately 400 gallons (1514.2-L) of water from the tank, and the tank was refilled with fresh well water on Monday mornings prior to creating the manure slurry treatment for that week.

### 2.3. Bacteriophage Cocktail Treatment

#### 2.3.1. Bacteriophage Assays to Determine Host Range

Phage spot assays were conducted to identify and develop a bacteriophage cocktail treatment that would be effective against the *Salmonella* serovars identified in the feedlot environment and cattle. A representative group of *Salmonella* isolates (n = 31) from a longitudinal study conducted in 2019 were selected to be evenly distributed across all variables of interest: 8 serovars, 7 months, 5 sample types, and 24 out of 30 pens (Appendix A) [18]. This provided a robust collection of isolates that were representative of the *Salmonella* population from the WTAMU Feedlot during 2019. These isolates from 2019 were used rather than the pre-trial isolates collected during this study at the WTAMU Feedlot because the development of the bacteriophage cocktail occurred concurrently with the pre-trial sampling period. Studies from our research group have found similar *Salmonella* serovars across several years at the WTAMU Feedlot [18,20,40].

The phages tested for the cocktail treatment originated from a study that investigated the *Salmonella* host range of 15 different *Salmonella*-infecting bacteriophages [41]. Three bacteriophages (Felix-O1, Melville, and Sw2) were selected. The routine test dilution (RTD), which is the dilution that results in countable plaques on the propagation host, was determined for the bacteriophage stocks. For this, each bacteriophage stock was serially diluted in SM buffer (1 M Tris-HCL (pH 7.5), NaCl, MgSO_4_, Gelatin (2%), dH_2_0) and spotted in duplicate on the corresponding propagation host, and the RTD was selected after overnight incubation at 37 °C. The RTD of each phage was then spotted in duplicate on the bacterial lawns of the 31 selected *Salmonella* isolates from the study in 2019 using the spot test methods previously described. *Salmonella* reference stains LT2-AMK (host for Felix-O1 and Melville) and FC1033C3 (host for Sw2) (American Type Culture Collection (ATCC), Manassas, VA, USA), which are known propagation hosts for the selected bacteriophages, were used as positive control comparison plates. The plates were scored based on plaque formation, or clearing, on the bacterial lawns due to cell lysis. Scoring compared the plaque formation on the test plates to the positive control plate, ranging from an equal number of plaques [+++], <50% of the plaques compared to the control plate [++], <10% of the plaques compared to the control plate [+], and no plaques/clearing [-]. Turbidity, signifying partial host cell death, was also noted by parenthesis. Phage selection for inclusion in the bacteriophage cocktail treatment was based on these host range results.

#### 2.3.2. Bacteriophage Cocktail Treatment Development and Application

Stocks of Melville and Sw2 phages were prepared individually in TSB to a final concentration of 2 to 6 × 10^10^ PFU/mL. The concentrated phage stocks were dispensed into 18 bottles (500 mL each) per bacteriophage and stored at 4 to 8 °C at the AgriLife Feedlot until use.

The phage cocktail treatment was applied on Day 1 and Day 15 of the experimental trial to 9 randomly selected pens. On application days, one bottle of bacteriophage Melville and one bottle of bacteriophage Sw2 were combined into the tank of a 4-gallon (15.1-L), 40 V battery powered backpack sprayer (Kobalt Tools, Mooresville, NC, USA) and diluted by filling the tank to the 4-gallon (15.1-L) mark with bottled spring water. One 4-gallon (15.1-L) backpack sprayer tank was used for a single feedlot pen (160 m^2^) for coverage of approximately 95 mL/m^2^. The application was applied evenly at an approximate rate of 0.5 GPM (1.9 LPM) for approximately 8 min per pen. The cattle remained in the pen during the application, which started from the front feedbunk and ended at the back fence. Prior to application, a 50 mL aliquot of the diluted bacteriophage from the backpack sprayer tank was collected and stored to determine bacteriophage viability and the final bacteriophage concentration.

### 2.4. Experimental Trial Study Design

#### 2.4.1. Pre-Trial Sampling—WTAMU Feedlot

Pre-trial sampling was conducted within 18 pens, dispersed throughout the WTAMU Feedlot in Canyon, TX, based on pen availability. A pen composite manure pack sample was collected from the pens a week prior (Day -43) to cattle arrival and used to represent the baseline *Salmonella* population at the WTAMU Feedlot. On Day -36, fecal grab, brisket swab, and rump swab samples were collected from 216 cattle, prior to allocating 12 cattle per pen based on body weight. The pen composite manure pack, fecal, rump swabs, and brisket swabs were also collected on Days -21 and -7 of the pre-trial period. Following sampling on Day -7, 10 randomly selected cattle per pen (n = 180) were transported to the AgriLife Feedlot for the experimental trial. The remaining 36 cattle at the WTAMU Feedlot were placed in one large pen to serve as a comparison group and were sampled monthly until slaughter. The experimental trial could not be conducted at the WTAMU Feedlot due to the lack of a dust abatement sprinkler system; therefore, the pre-trial period was used to establish enteric and hide *Salmonella* populations that would likely be similar serovars to those isolated in 2019, prior to relocating the cattle to the AgriLife Feedlot.

#### 2.4.2. Experimental Trial—AgriLife Feedlot

A group of 18 pens with pre-installed dust abatement sprinklers were utilized at the AgriLife Feedlot in Bushland, TX, for the trial. Pens 1 to 3 and 16 to 18 were soil based and pens 4 to 15 were fly ash based. The pen composite manure pack samples were collected a week prior to (Day -14) the cattle’s arrival and the day (Day -7) that the cattle arrived at the AgriLife Feedlot to determine the presence of *Salmonella* in the environment. The cattle were placed in 10-head pens, maintaining previous allocations from the WTAMU Feedlot. There was a week-long acclimation period prior to the start of the trial. Trial sampling occurred weekly on Days 0 (baseline), 7, 14, and 21 (Figure 1). On sampling days, a total of 552 samples were collected, consisting of one pen composite manure pack sample from each pen (n = 18), a fecal grab sample (n = 178), a brisket hide swab (n = 178), and a rump hide swab (n = 178) from each animal (2 cattle died due to reasons unrelated to this experiment, and therefore, their samples were not collected).

Due to the unbalanced design, there were 4 pens (2, 4, 7, and 9) that received only the manure slurry treatment, 4 pens (10, 13, 15, and 18) that received only the phage cocktail treatment, 5 pens (1, 3, 5, 6, and 8) that received the combination treatment, and 5 pens (11, 12, 14, 16, and 17) that served as control pens and received no treatment (Figure 2). After completion of the experimental trial, the cattle remained in their pens for monthly follow-up sampling until slaughter.

#### 2.4.3. Follow-Up Sampling—WTAMU and AgriLife Feedlots

The cattle at the AgriLife Feedlot were sampled monthly, from September 2021 to January 2022. Sampling in September and October included the composite environment manure pack, fecal grabs, and rump swabs due to an on-going study using the same cattle. A reduction in *Salmonella* prevalence was observed, resulting in the decision to only collect composite environment manure pack and fecal grabs from November to January. Cattle fecal grabs and the composite environment manure pack were also collected monthly, from August 2021 to February 2022, at the WTAMU Feedlot. All cattle from both feedlots were transported for slaughter upon reaching market weight. Subiliac lymph nodes were collected from beef carcasses during the harvest process.

### 2.5. Sample Collection

On sampling days, the cattle were brought by pen to the animal handling facility and restrained in a squeeze chute to facilitate fecal and hide sampling. Composite environment manure pack samples were collected after the completion of cattle sampling.

For the hide swab samples, Whirl Pak^®^ Speci-Sponge^®^ bags (Whirl-Pak by Nasco, Fort Atkinson, WI, USA) were filled with 25 mL of Butterfield’s phosphate buffer (Hardy Diagnostics, Santa Maria, CA, USA) the night prior to sample collection to pre-moisten the sponges. Separate sponges were rubbed across a 1 m^2^ area of the brisket and rump of each animal. Approximately 25 g of fecal sample was collected by rectal grab using obstetric sleeves. The obstetric sleeves were inverted, knotted, and placed in a Whirl Pak^®^ bag (Whirl-Pak by Nasco, Fort Atkinson, WI, USA) for secondary confinement. Approximately 25 g of composite environment manure pack sample was collected using a plastic spoon from six different locations across the diagonal of the pen starting from the feedbunk and ending at the back of the pen and placed into a Whirl Pak^®^ bag (Whirl-Pak by Nasco, Fort Atkinson, WI, USA). All samples were kept on ice and shipped overnight or driven back the day following sampling to the Texas A&M University School of Veterinary Medicine & Biomedical Sciences (TAMU-VMBS) in College Station, TX.

The first group of cattle (n = 14) from the WTAMU Feedlot was shipped on Day 142 for slaughter. All of the cattle (n = 177) from the AgriLife Feedlot were shipped on Day 178 for slaughter. The second group of cattle (n = 13) from the WTAMU Feedlot was shipped on Day 211 for slaughter. Left and right subiliac lymph nodes were collected warm, prior to carcass chilling. Per carcass, the left and right lymph nodes were pooled into a single sterile sample bag, generating one sample per carcass. Terminal fecal grab samples were collected from all cattle the day of or the day prior to slaughter.

### 2.6. Salmonella Selective Enrichment and Isolation

The hide swabs were processed upon arrival to the laboratory by adding 75 mL of pre-measured TSB to each Whirl Pak^®^ Speci-Sponge^®^ bag, which was hand massaged 10 times and incubated for 3 h at 42 °C prior to continuing with the selective enrichment protocol for *Salmonella*. All other sample types were processed, selectively enriched, and confirmed as *Salmonella* using the same methods as previously described [18].

### 2.7. Whole Genome Sequencing

A selection of isolates (n = 64) from the pre-trial sampling Days -36 and -21 (as previously discussed) and all isolates from the experimental trial (n = 557) underwent DNA extraction and whole genome sequencing as previously detailed [18]. The only modifications included the use of an xGen™ EZ Library Prep Kit, an xGen™ Deceleration Module, xGen™ Normalase UDI Primers, and an xGen™ Normalase Module (IDT Corporation, Newark, NJ, USA) for library preparation. The xGen™ Deceleration Module reagent was used in place of one reagent in the xGen™ EZ Library Prep Kit during the enzymatic fragmentation step to slow down the reaction and increase the fragment size (~500 bp) and consistency of the library. Multiplexed DNA libraries were loaded 32 at a time with a concentration of 13.5 pM on an Illumina MiSeq^®^ platform (Illumina, San Diego, CA, USA).

Additionally, pen surface (n = 4), fecal grab (n = 23), brisket swab (n = 23), and rump swab (n = 4) isolates from pre-trial sampling Day -7 and all follow-up sampling isolates (n = 43) were shipped to the Advanced Molecular Detection team at the Texas Department of State Health Services (DSHS) in Austin, TX, for whole genome sequencing.

### 2.8. Data Analysis

All bioinformatic analyses for *Salmonella* and serovar prevalence were conducted using methods as previously described [18]. Briefly, raw reads were trimmed, and assemblies were evaluated for quality control purposes. *Salmonella* serovars and AMR genes were identified from assemblies. A phylogenetic tree was constructed with isolates (n = 611) from the trial, including 54 isolates from Day -7 and 557 isolates from Days 0 to 21.

Statistical analyses of *Salmonella* prevalence from the sample culturing results and serovar prevalence from the isolate sequencing results were conducted using Stata version 17.0 (StataCorp, College Station, TX, USA). A multilevel mixed effects logistic regression model was used to evaluate the impact of treatment on *Salmonella* prevalence by sample type (individual cattle fecal or hide swabs). The feedlot pen was used as a random effects variable to account for clustering by pen. A 3-way full factorial term with the manure slurry treatment, phage cocktail treatment, and collection day variables was included in the model, evaluating all 4 combinations of treatments across all 5 collection days. A pen environment interaction term with day was included as a fixed effect, representing the impact that *Salmonella* from the pen environment had on the cattle housed within the pen. The pen base material (soil based or fly ash based) was initially included in the model but did not provide significant contribution to the variance and was excluded from the model. Predictive margins for *Salmonella* prevalence were generated, including the 3-way full-factorial term, and margin plots were created for each sample type. Multiple correspondence analysis (MCA) was used to assess associations between treatment, day, and sample type with *Salmonella* serovars. The multiple correspondence analysis plot was created with a Burt’s matrix approach using principal normalization to portray the relationships between the *Salmonella* serovar and treatment group. The serovars Edinburgh and 61:l,v:1,5 were excluded because of low frequency. Only two dimensions were used because there were negligible contributions to the inertia past dimension two.

## 3. Results

### 3.1. Pen Environment Selection for the Manure Slury Treatment

Of the 64 pre-trial isolates that were sequenced to determine feedlot pen environments with optimal *Salmonella* populations for the manure slurry treatment, there were nine serovars identified; the serovars Virginia (26.6%, 17/64), Cerro (23.4%, 15/64), and Montevideo (20.3%, 13/64) were identified most frequently (Figure 3). In a previous study at the WTAMU Feedlot, the serovars Anatum, Kentucky, and Lubbock were commonly identified in cattle lymph nodes; therefore, we excluded Pens 2, 4, 18, 19, 20, 48, 51, and 57 from consideration to avoid those serovars [18]. Pens 1 and 51 were also excluded from consideration because AMR genes were identified in isolates within these pens. Specifically, the *cat* and *tet*(J) genes were identified in a composite environment manure pack isolate from Pen 1, potentially conferring resistance to chloramphenicol, doxycycline, and tetracycline. In Pen 51, the *tet*(A) gene, potentially conferring resistance to doxycycline and tetracycline, was identified in a fecal isolate. Pens 52, 53, 54, and 55 had low *Salmonella* prevalence in cattle feces and were the only pens with no *Salmonella* identified in the composite environment manure pack on both Days -36 and -21, except for Pen 57 which was previously excluded (the serovar Anatum identified in cattle feces). The *Salmonella* that were identified in cattle feces from Pens 52 to 55 were pan-susceptible serovars (Cerro 5/10, Virginia 3/10, Montevideo 1/10, and Agona 1/10) that are typically found in cattle and the feedlot environment. Based on these results, the composite environment manure pack from Pens 52 to 55 were selected for use in the manure slurry treatment.

Naturally occurring phages were identified retrospectively, approximately two months after collection, in the environment of Pens 52 to 55 using phage spot test assays. Pens 54 and 55 displayed the broadest host range from the selectively enriched virome. Interestingly, the virome from all pens resulted in complete lysis of the *Salmonella* serovars Montevideo and Cerro, and a high degree of lysis for the serovar Virginia. A lower level of clearing was observed for the serovars Newport and Muenster, primarily from the virome in Pens 54 and 55. All other *Salmonella* serovars that were tested were not susceptible to the natural phages from any of the selected pens (Figure 4).

### 3.2. Bacteriophage Selection for the Bacteriophage Cocktail Treatment

The RTD phage assay results were used to determine which phage to include in the cocktail (Figure 5). Phage Felix-O1 showed inefficient clearing of the serovars Virginia and Cerro and no clearing for Lubbock, Anatum, and Kentucky. Phage Melville had the broadest host range and highest degree of bacterial clearing of the three phages tested. Phage Sw2 was highly effective at lysing the *Salmonella* serovar Anatum. Retrospective testing was completed and determined that the serovar Muenster was not a suitable host for Melville, Felix, or Sw2. Phage Melville lysed the serovar Newport to a limited degree.

High titer stocks of phages Melville and Sw2 were prepared and were still viable during the cocktail application on Day 1 and Day 15 (Table 1).

### 3.3. Pre-Trial Results

There were a total of 1991 samples collected from Days -43 to -7 during the pre-trial period (72 composite environment manure packs, 641 fecal grabs, 639 rump swabs, and 639 brisket swabs). Brisket and rump swabs from two cattle on Day -7 were excluded due to mislabeling. There were three cattle that were culled and five that died during the pre-trial study period (one prior to Day -21 sample collection, six prior to Day -7 sample collection, and one on Day -7 after sample collection); therefore, only 28 cattle remained at the WTAMU Feedlot for the follow-up period.

The composite environment manure pack *Salmonella* prevalence at the WTAMU Feedlot on Day -43 prior to cattle placement was 27.8% (5/18). *Salmonella* prevalence continued to increase throughout the pre-trial period (Day -36: 38.9%, 7/18; Day -21: 66.7%, 12/18; and Day -7: 77.8%, 14/18; Appendix A).

The *Salmonella* pre-trial baseline (Day -36) prevalence for cattle at the WTAMU Feedlot was 19.9% (43/216) for fecal grab samples and 77.3% (167/215) and 81.9% (177/215) for cattle rump and brisket hide samples, respectively. There was a decline in prevalence on Day -21 observed in all cattle sample types. On the last pre-trial sampling day, Day -7, the highest prevalence at WTAMU throughout the entire study was observed in feces (36.7%, 77/210), rump (80.3%, 167/208), and brisket (81.7%, 170/208) samples. The whole genome sequencing results for the pre-trial Days -36 and -21 were previously discussed.

### 3.4. Experimental Trial Results

Of the 180 cattle selected for the trial, 2 died during the acclimation period, resulting in 178 cattle included in the experimental trial sampling. There were 13 fecal samples that were unable to be collected on Day 0. This resulted in a total of 2195 samples collected during trial Days 0 to 21 (72 composite environment manure packs, 699 fecal grabs, 712 rump swabs, and 712 brisket swabs).

#### 3.4.1. Pre-Treatment *Salmonella* Prevalence

The subset of cattle samples collected on the day of arrival (Day -7) to the AgriLife Feedlot, which will be referred to as the incoming prevalence, had higher *Salmonella* prevalence compared to all other days throughout the trial. On Day -7, cattle hide (brisket: 78.7%, 140/178; rump: 80.9%, 144/178) prevalence was higher than fecal prevalence (38.2%, 68/178; Appendix A).

There were no *Salmonella* identified in any pen composite environment manure pack samples from the AgriLife Feedlot on Days -14 and -7. After a week-long acclimation period in a *Salmonella*-naïve environment, prevalence decreased on Day 0 in brisket, rump, and fecal samples (42.7%, 76/178; 22.5%, 40/178; and 12.1%, 20/165, respectively) compared to the incoming prevalence (Appendix A). As expected, composite environment manure pack prevalence increased from 0% (0/18) on Day -7 to 22.2% (4/18) on Day 0 (baseline prevalence).

#### 3.4.2. Post-Treatment *Salmonella* Prevalence

After the initiation of treatment, the average *Salmonella* prevalence in feces, brisket, and rump samples was lower in all treatment groups compared to the control group (Appendix A). The lowest post-treatment prevalence (Days 7 to 21) in both brisket (18.4%, 27/147) and rump samples (15.6%, 23/147) was observed in combination treatment pens. Conversely, the average *Salmonella* prevalence was lowest in fecal grabs (12.5%, 15/120) and composite environment manure pack (58.3%, 7/12) samples from the phage cocktail treatment pens.

A multi-level mixed effects logistic regression model was used to evaluate the significance of treatment on the *Salmonella* prevalence of each sample type (Appendix A), with the predicted margins of *Salmonella* prevalence for treatment groups by collection day presented in Figure 6. There was little change in *Salmonella* prevalence in the fecal samples post treatment (Figure 6a). Important post-treatment trends can be observed for the brisket swab samples (Figure 6b); for the manure slurry treatment group, there was an initial increase in *Salmonella* followed by a steady decrease, though the difference was not significant, and for the phage cocktail treatment group, there was an increase in prevalence between the two application days though the difference was not significant. Within the rump swabs (Figure 6c), the *Salmonella* prevalence was significantly lower in the combination treatment group compared to the control group on Days 14 and 21.

### 3.5. Whole Genome Sequencing

#### 3.5.1. Quality Metrics

The pre-trial isolates (n = 64) were sequenced prior to the trial isolates using materials and methods previously described [18]. The average sequencing coverage depth for the pre-trial isolates was 32X. The average genome size was 4,796,291 bp, the average GC content was 52%, and there was an average of 144 contigs per assembly. The experimental trial isolates (n = 557) using updated materials and an increased loading concentration as described in the Methods section resulted in an average coverage depth of 45X. The average genome size of these isolates was 4,758,125 bp, with an average GC content of 52% and an average of 43 contigs per assembly.

#### 3.5.2. *Salmonella* Serovar Composition

There were 11 different serovars identified from Day -7 (n = 54) and throughout the trial (n = 557), with the majority being Montevideo (42.6%, 260/611). Individual serovar trends by collection day, sample type, feedlot pen, and individual cattle can be viewed in Figure 7.

The serovar composition from post-treatment fecal and hide within the phage cocktail treatment group was similar to the control group, with Montevideo (44.3%, 31/70; 68.6%, 107/156) and Virginia (22.9%, 16/70; 22.8%, 38/167) most frequently identified, respectively.

Within the manure slurry treatment group, the *Salmonella* serovar Newport was most common (46.4%, 32/69) and was almost entirely observed in Pen 9. Although the serovar Newport was not identified until Day 7 after treatment started, we did not identify the serovar Newport in the composite environment manure pack (Pens 52 to 55) that was used for the manure slurry treatment. The *Salmonella* serovar Cerro was also more commonly identified in the manure slurry group, especially within cattle feces, compared to the other treatment groups. Unsurprisingly, Cerro was identified in the composite environment manure pack (Pens 52 to 55) used for the manure slurry treatment (Figure 3).

The combination treatment group had an entirely different serovar composition and the highest serovar diversity, with the serovar Muenster being the most abundant (43.2%, 32/74). Interestingly, no *Salmonella* Montevideo isolates were observed in the composite environment manure pack for the combination treatment group. The Serovar Anatum, most frequently identified in the cattle brisket hide samples, was commonly identified in Pen 3, and the serovar Virginia was identified in Pen 6. *Salmonella* pen dynamics can be observed over time in Pen 8 of the combination group; on Day -7, only the serovar Kentucky was identified, and from Days 0 to 21, you can observe the continual reduction in Kentucky and increase in the serovar Muenster.

The multiple correspondence analysis (MCA) plot, used to assess associations between the serovar and treatment group, had a total inertia of 56.9% (Figure 8). Because of the long distance from the origin and small angles, Muenster and Kentucky are distinctly different from other serovars in that they were frequently observed in samples originating from the combination treatment pens. For similar reasons, the serovars Newport and Cerro were commonly identified in pens that received the manure slurry treatment; however, this relationship was not as pronounced. As the phage cocktail treatment and control groups are closer to the origin, it signifies that these treatment groups are less distinct from each other in terms of serovar composition. The serovars Montevideo and Lubbock are strongly associated with both the phage cocktail treatment and control groups. The serovar Virginia was frequently observed in the control, manure slurry, and phage cocktail groups; however, the relationship was weaker than for the serovars Montevideo and Lubbock. The serovars Anatum and Lille had no obvious relationships to specific treatment groups. An MCA plot was created to evaluate associations between serovar and sample type, but the only relationship observed was that Anatum was associated with brisket hide samples.

### 3.6. Phylogenetic Analysis

The phylogenetic tree in Figure 9 contains a random selection of isolates sequenced from Day -7 (n = 54) and all isolates recovered from trial Days 0 to 21 (n = 557). Serovar trends by treatment group can be identified in the tree; for example, the majority of the Montevideo isolates are from the control group (represented as orange in the treatment ring), as previously stated. The *Salmonella* serovar is also highly conserved by the feedlot pen; *Salmonella* serovar Montevideo isolates are mainly clustered into two pen groups (Pens 10 to 12: green shades and Pens 13 to 15: teal shades), as are *Salmonella* serovar Virginia isolates (Pens 4 to 6: orange shades and Pens 16 to 18: blue shades). The two clusters of Virginia isolates also correspond to different treatment applications (control group: orange; combination group: bright green). The serovars Muenster, Kentucky, and Newport were most frequently identified in Pens 7 to 9 (yellow shades). The Serovar Anatum was identified across pens, rather than a specific pen block, and all other serovars were too few to detect a pen effect. A phylogenetic tree of all sequenced isolates (n = 717), including isolates from Days -36, -21, and the follow-up months showed similar clustering patterns; however, the clusters were less distinct (Appendix A).

### 3.7. Antimicrobial Resistance Genes

Antimicrobial resistance (AMR) genes in the pre-trial (Day -7) isolates were previously discussed. For the trial (Days 0 to 21) isolates, AMR genes were only identified in 2.2% (12/557) of the isolates. Within the serovar, the AMR profiles were consistent; all serovar Lille isolates harbored the *dfr*A12, *aad*A2, and *sul*3 resistance genes, and all serovar Lubbock isolates harbored the *tet*(A) gene. Interestingly, the Lille isolates all originated from different pens, but the Lubbock isolates originated from Pens 12 and 13. One Anatum isolate had additional AMR genes that confer resistance to phenicols (*flo*R), aminoglycosides (*aph(6)-ld*, *aph(3′)-lb*), tetracyclines (*tet*(A)), sulfonamides (*sul*2), and ß-lactams (*bla*-_TEM-lA_). The isolates were not tested for phenotypic resistance or reduced susceptibility.

### 3.8. Follow-Up Results

There were 28 cattle that remained in pen NO2 at the WTAMU Feedlot for follow-up sampling, and the *Salmonella* prevalence in cattle feces ranged from a high of 85.7% in September to a low of 57% in November (Table 2). Terminal fecal samples were not available in December due to a collection error. One steer died due to causes unrelated to the experiment during the follow-up period in January at the AgriLife Feedlot. The *Salmonella* prevalence in cattle feces at the AgriLife Feedlot was lower than at the WTAMU Feedlot during the follow-up period and ranged from a high of 7.9% in September to a low of 0.6% in January (Table 2).

There were six *Salmonella* serovars (Anatum 9.3%, 4/43; Kentucky 2.3%, 1/43; Lille 4.7%, 2/43; Montevideo 14.0%, 6/43; Muenster 48.8%, 21/43; and Virginia 20.9%, 9/43) identified from the feces, rump, and composite environment manure pack isolates that were from samples collected during the follow-up period at the AgriLife Feedlot. The majority of the isolates originated from control pens (39.5%, 17/43) and combination treatment pens (32.6%, 14/43). Virginia was observed solely in the control pens (52.9%, 9/17), as well as the majority of the Montevideo isolates (29.4%, 5/17). The Serovar Muenster was mainly identified in Pen 8, which had received the combination treatment (57.1%, 12/21), and Pen 9, which had received the manure slurry treatment (33.3%, 7/21).

### 3.9. Cattle Subiliac Lymph Nodes

In addition to the eight cattle that either died or were culled, one carcass from the WTAMU Feedlot did not pass initial USDA inspection and was separated for decontamination, and therefore, its subiliac lymph nodes were inaccessible. A total of 54 (27 pooled samples) left and right subiliac lymph nodes were collected. The overall *Salmonella* prevalence observed in the cattle subiliac lymph nodes from the WTAMU Feedlot was 55.6% (15/27), with a prevalence rate of 42.9% (6/14) on Day 142 and 69.2% (9/13) on Day 211 (Table 2).

All the cattle from the AgriLife Feedlot went to slaughter together on Day 178, in-between the two WTAMU Feedlot slaughter dates. There was one additional steer that died prior to slaughter, and its lymph nodes were not available for collection, resulting in 352 (176 pooled samples) lymph nodes collected. Surprisingly, there was only one pooled lymph node sample in which *Salmonella* was identified (0.57%, 1/176).

## 4. Discussion

This experiment assessed the utility of two different treatments applied in the beef cattle feedlot environment as *Salmonella* mitigation techniques. The manure slurry was intended to increase the presence of typical pen flora including any resident phages. The phage cocktail treatment was designed to infect *Salmonella* serovars previously identified at the WTAMU Feedlot [18]. Across the entire study, composite environment manure pack, cattle feces, brisket swabs, and rump swab samples were collected to observe changes in *Salmonella* prevalence and serovar composition between two research feedlots (i.e., the WTAMU Feedlot and the AgriLife Feedlot) and the four treatment groups (i.e., manure slurry, phage cocktail, combination of the two, and negative control). Cattle subiliac lymph nodes were also harvested at slaughter to determine the effect of the treatments on the prevalence of *Salmonella* harbored in lymph nodes. This trial was conducted under the premise that these pre-harvest environmental treatments would reduce *Salmonella* on the pen surface, thereby reducing *Salmonella* carriage in the cattle feces and lymph nodes. A reduction in *Salmonella* prevalence was observed in all samples (i.e., feces and hides) except for the lymph nodes (which could not be analyzed due to low *Salmonella* prevalence in all treatment groups) and on the pen surface (due to an underpowered sample size). Results from this research suggest that phage treatments applied to the feedlot environment may be effective at reducing *Salmonella*, especially on the hides of cattle.

The manure slurry treatment was meant to represent pen water run-off that is held in retention ponds at feedlots and is typically used for irrigation and dust-abatement purposes. Although Purdy et al., identified several *Salmonella* serovars present in feedyard playas throughout the year, Loneragan and Brashears, determined that the use of retention pond water has no effect on pathogen carriage in animal feces and hide [42,43]. Their research suggests there may be another component within retention pond water keeping pathogen populations at bay, such as bacteriophages. The rationale behind the manure slurry treatment is further supported by studies that have identified *Salmonella*-infecting phages from feces, feed, water, and pen surface soil samples from Texas feedlots and New York dairy farms [38,39]. Phages from these studies were identified in samples with intermediate or low *Salmonella* prevalence, suggesting that phages can be recovered regardless of *Salmonella* being present [38,39]. This previous research supported our decision to select the environment from the surface of feedlot pens that had low *Salmonella* prevalence for the manure slurry treatment [38,39].

The initial increase in *Salmonella* prevalence in the environment of the manure slurry treatment pens was anticipated, and we expected to observe serovars that originated from the composite environment manure pack used to create the manure slurry at a higher frequency compared to other serovars. Although this was true for most serovars, the serovar Agona was not recovered in any of the manure slurry-treated pens, even though it may have been included in the manure slurry. The naturally occurring phages present in the manure slurry may have contributed to the observed decrease in specific serovars in the manure slurry treatment group. However, the concentration of the endogenous phages in the applied water was not determined. This method of applying phages through sprinkler water has not been explored in the cattle industry. Typically, phages are administered pre-harvest to agricultural animals, most commonly poultry and swine, orally or through treated drinking water [44]. The results from Sheng et al. demonstrated a significant reduction (*p* < 0.05) in *E. coli* O157:H7 in cattle that received phages through drinking water and an anal swab [34]. However, no studies have investigated the environmental administration methods we present here to mitigate *Salmonella* in beef cattle.

Commercial bacteriophage cocktail products, such as PhageGuard S^™^, are available to be sprayed topically on food or decontaminate contact surfaces in the poultry industry for *Salmonella* mitigation [45]. Commercial bacteriophage products are typically sprayed on post-harvest products and processing facilities, not used in the pre-harvest setting [46]. The results from these products cannot be generalized to the pre-harvest setting of the beef industry because the feedlot environment differs greatly from poultry settings and *Salmonella* serovars are host adapted [47]. In our study, we curated a phage cocktail specific to the identified *Salmonella* populations at the feedlot. The *Salmonella* prevalence for the phage cocktail treatment suggests the treatment was able to keep *Salmonella* populations static over time or there was no effect. Importantly, there were fewer *Salmonella* identified on collection Day 7 which was collected 7 days after the last phage cocktail treatment application in comparison to collection Day 21 which was 14 days after the last phage cocktail treatment application. This suggests the bacteriophage cocktail may need to be applied more frequently or that phages may become inactive over time. With our study design, we were unable to explore the optimal duration and frequency of environment phage treatment applications. The persistence of phages is dependent on their individual morphology, external temperature, and environmental composition (i.e., pH, salinity, and ions) [48]. Furthermore, the reduction in water content within the complex matrix of the pen surface between treatments could reduce the ability of phages to migrate to bacterial hosts [49,50]. The results of this study provide evidence that the combined effect of both treatments was superior to individual treatments. The additional moisture added to the environment from the manure slurry may have contributed to the combination treatment being more effective. The combination treatment should also have a broader *Salmonella* host range, providing a greater impact on diverse *Salmonella* populations. To our knowledge, there are no published studies investigating the combination of natural phage and targeted phage cocktail applications for pathogen mitigation in the beef industry.

We recognize that the complete removal of *Salmonella* from the feedlot environment is not realistic, so the primary goal of the phage treatments was to shift to non-pathogenic *Salmonella* serovars. The presence of AMR *Salmonella* in this study were too infrequent to determine the significance of these occurrences. Nonetheless, the AMR profiles that we observed were consistent within serovars. This has been observed in previous work conducted at the WTAMU Research Feedlot, which showed that *Salmonella* serovar Reading isolates had a consistent AMR profile (ACSSuT: resistant to ampicillin, chloramphenicol, streptomycin, sulphonamides, and tetracycline) [40]. As AMR profiles are typically associated with particular *Salmonella* serovars, this suggests that phage selection should be focused on serovars of concern. The serovar Newport has been identified as an issue for the safety of beef products. From 2016 to 2019, there have been over 540 illnesses and 160 hospitalizations linked to *Salmonella* Newport-contaminated ground beef [51,52]. In this study, *Salmonella* Newport was identified at the AgriLife Feedlot after the start of treatment, mainly within the manure slurry group. However, we were unable to identify any evidence suggesting it was spread via manure slurry treatment. The Newport serovar at the AgriLife Feedlot may have originated via contact transmission from other cattle at the feedlot that interacted with personnel or equipment used to provide feed, but the source remains unknown [53,54]. Importantly, we determined that the naturally occurring phages in the treatment were capable of lysing the serovar Newport which decreased in prevalence by the end of the trial.

The majority of carcass contamination comes from bacteria located on the cattle hide, making it an important mitigation target [7,55]. According to previous research, the cattle rump tends to have a *Salmonella* prevalence rate ranging from 21–90%, which is typically lower compared to the brisket region ranging from 75–98% [7,56]. The *Salmonella* prevalence in our study had a greater decrease within rump swab samples from the manure slurry and combination treatment groups than within the brisket swab samples. These results were anticipated because the rump samples were expected to represent aerosolized treatments applied by the sprinklers and also aerosolized pen surface material that potentially settled on the dorsal region of the cattle. The results in our study could also be attributed to the treatment washing the *Salmonella* off the hide and should be compared with plain water treatment in future studies. The brisket samples were meant to be representative of cattle interactions with the pen environment, suggesting that the phage cocktail treatment should have had a greater impact on brisket *Salmonella* populations, although this was not the case. Unlike the manure slurry, the phage cocktail was not directly sprayed onto the cattle hide; therefore, the phage cocktail treatment may need to be applied more frequently or at a higher dose in the environment for continual reductions in *Salmonella*. Mitigating *Salmonella* with a treatment applied to the cattle hide and the environment is desirable because of the natural interactions between cattle and the environment.

We hypothesized that through cattle grooming behaviors, bacteriophages on cattle hides may be ingested and arrive in the gastrointestinal tract of cattle, thus impacting fecal *Salmonella* populations through the fecal–oral route [57]. Although the *Salmonella* prevalence in cattle feces was lower in the combination and manure slurry groups by the end of the trial compared to the control group, it was less pronounced compared to the hide samples. These results suggest that this transmission route may not supply a high enough concentration of phages to have a significant effect on *Salmonella* in feces. Additionally, phages may become inactivated due to the pH changes in the digestive system or from the complex matrix of cattle feces which contains a diverse microbiota, water, and dry matter, which is composed of nitrogen-free extract, acid detergent fiber, and crude fiber [58,59,60]. There was also little change in *Salmonella* prevalence within the composite environment manure pack over the course of the trial. Similar results have been observed in laboratory soil models, which determined that phages were able to reduce *Salmonella* in feedlot pen surface soil (0.53 to 1.38 log_10_ CFU/g) and hide (0.50 to 1.75 log_10_ CFU/cm^2^), however less effectively in feedlot pen surface soil [61]. This study suggested that phages may not come into contact with *Salmonella* in complex matrices such as feedlot pen surface soil which is why the treatments were more effective on cattle hide. We did not conduct testing on *Salmonella* resistance to phages; however, a previous study reported that 24 hrs after treatment, there were no phage-resistant *Salmonella* identified in feedlot soil [61].

Pen effects were evident in our analyses, suggesting that these effects needed to be included in our statistical model. Apart from treatment applications, pen effects could be attributed to geographical location, whether they are upwind or downwind of treatment pens, or based on incoming cattle *Salmonella* populations. Similar feedlot pen effects within serovars have been observed previously [20]. With the inclusion of pre-trial and follow-up isolates, pen-level trends were not as evident (Appendix A). Unsurprisingly, this suggests that *Salmonella* observed over shorter periods of time are more alike. Another consideration is that these changes are because the WTAMU Feedlot contributes to higher *Salmonella* diversity from sustained *Salmonella* populations that have had the opportunity to evolve over time compared to the AgriLife Feedlot. However, similar *Salmonella* serovars have been observed at the WTAMU Feedlot during research studies that occurred across several years [18,20,40].

Rain or temperature changes are potential reasons for the *Salmonella* prevalence fluctuations observed across the course of the study. There were no drastic fluctuations in *Salmonella* prevalence amid weather events during the experimental trial that occurred concurrently across all sample types, with the exception of cattle samples on trial Day 21. However, if weather did influence the drop in *Salmonella* prevalence in samples from Day 21, it was compounded by the treatments. During the follow-up period, the *Salmonella* prevalence decreased across the collection months as the temperatures decreased, which was expected as previous research has demonstrated a seasonality effect on *Salmonella* prevalence [62].

Despite the two feedlots having the same climate and similar weather conditions (located approximately 30 miles apart), differences in prevalence across feedlots were apparent during the follow-up period and in cattle subiliac lymph nodes but are unlikely to be related to the treatments. New cohorts of cattle are consistently introduced to the WTAMU Feedlot, but the AgriLife Feedlot pens were empty for approximately 5 months prior to cattle arrival. Cattle are known to shed *Salmonella* in their feces to the pen environment, but there are a lack of studies researching *Salmonella* dynamics in the feedlot environment [63]. Because of this, the duration of *Salmonella* survival in the environment without the continual reintroduction of *Salmonella* populations is unknown. There have been studies that have investigated the presence of *Salmonella* in treated manure or *Salmonella*-inoculated soil and the growth of pathogens on crops which suggested that *Salmonella* can persist anywhere from 120 to 231 days in the soil [64,65]. In fact, the USDA requires there be at least 120 days between manure applications and the harvest of crops, suggesting that pathogens are persistent within the environment during that time [66]. The AgriLife Feedlot pen dormancy exceeded that length of time; however, we expected that the pre-trial period at the WTAMU Feedlot would be sufficient to develop robust *Salmonella* populations within the cattle that could be transferred and maintained at the AgriLife Feedlot. Additionally, a previous longitudinal study determined that *Salmonella* identified in cattle lymph nodes were more often related to *Salmonella* from pens that cattle were housed in for a longer duration of time [18]. This supports the theory that the higher observed prevalence in cattle lymph nodes at the WTAMU Feedlot was due to the consistent presence of *Salmonella* in the pen environment which resulted in sustained cattle colonization. We did not anticipate these feedlot differences, which hindered our ability to observe the impact of different bacteriophage treatments on *Salmonella* harbored in cattle lymph nodes.

Despite the low observed prevalence during the follow-up period at the AgriLife Feedlot, there are observations that suggest the bacteriophage may have continued to mitigate *Salmonella* after applications ceased. The highest prevalence during the follow-up period was observed in the control pens, suggesting there may be long-lasting effects from the bacteriophage treatments. The serovars that were observed during the follow-up period at the AgriLife Feedlot were consistent with what was observed during the experimental trial, and pen-level effects remained. These results suggest that the bacteriophage treatments may have long-lasting effects for *Salmonella* mitigation; however, this conclusion is limited by the low prevalence and sample size observed during the follow-up period.

We were also limited to observing reductions in *Salmonella* prevalence rather than concentration in our study. *Salmonella* concentrations were unable to be determined because competing bacterial growth inhibited our ability to decipher accurate *Salmonella* counts. This limits our understanding of the true impact of treatment because repeated samples that remained *Salmonella* positive throughout the study may have had reduced concentrations of *Salmonella* that were not observed. Because of this, the effect of treatment may be underrepresented, thus necessitating further research. Apart from the study limitations, there are practical implications to be considered by feedlots interested in these customizable pre-harvest *Salmonella* mitigation strategies.

Each of the methods presented in our study do require prior knowledge of the *Salmonella* populations present in the feedlot environment and feedlots interested in using these methods would need to conduct initial testing to identify *Salmonella* serovars, if not already known. Phage cocktails are more time intensive to test, develop, and apply but provide a higher level of control and precision. A phage cocktail treatment can be used to target specific serovars of concern or address multiple serovars by using a broad host range phage. The development and application of the manure slurry can be carried out easily at feedlots that have a sprinkler system already installed. The manure slurry treatment can be generalized for shifting whole populations towards pan-susceptible *Salmonella.* These treatments are desirable for use at agricultural facilities because they are environmentally sustainable, non-invasive to animals, and can be developed according to the feedlots’ pathogen risk needs.

## 5. Conclusions

We determined that bacteriophage treatments applied to the feedlot environment can reduce *Salmonella* on cattle hide and in cattle feces. The manure slurry treatment was the more effective of the two singular treatments, and the combination treatment was the only treatment that significantly reduced *Salmonella*, which was especially effective at reducing *Salmonella* on cattle hide. Both bacteriophage treatments investigated in this study were safe and easy to use, making them promising *Salmonella* mitigation techniques that could be implemented in the pre-harvest setting.

Future studies involving feedlots known to have high *Salmonella* prevalence will help determine if phages are effective at reducing *Salmonella* in the feedlot pen environment and bovine lymph nodes. Additionally, studies testing the increased frequency or duration of phage applications will help with the optimization of treatment parameters.

## Figures and Tables

**Figure 1 animals-13-03170-f001:**
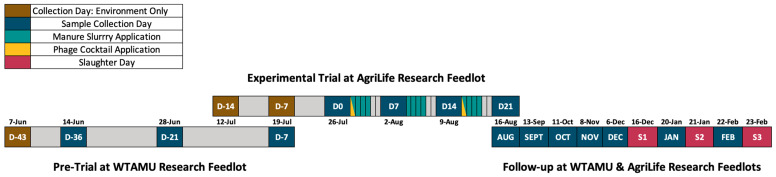
Baseline composite environment manure pack collection days (D#: brown), collection days (D#: dark blue), manure slurry application days (teal), phage cocktail application days (yellow), and slaughter days (S#: red) are indicated for the pre-trial, trial, and follow-up periods. The distances between events in the follow-up period are condensed and do not accurately represent time.

**Figure 2 animals-13-03170-f002:**
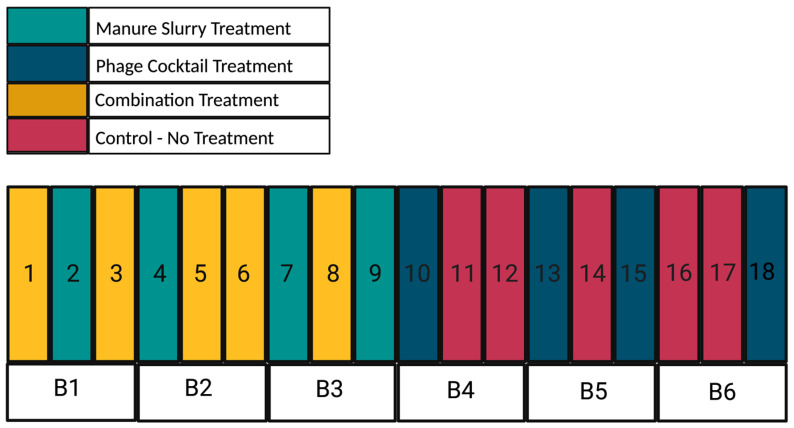
Layout of pens (1–18), blocked (B1–B6) based on the sprinkler system, at the AgriLife Feedlot. Treatments were applied in a 2 × 2 full factorial unbalanced design resulting in 4 treatment groups: (1) manure slurry (teal), (2) phage cocktail (dark blue), (3) combination (yellow), and (4) control (red). Block 1 (Pens 1 to 3) and Block 6 (Pens 16 to 18) were soil based pens and Blocks 2 to 5 (Pens 4 to 15) were fly ash based pens.

**Figure 3 animals-13-03170-f003:**
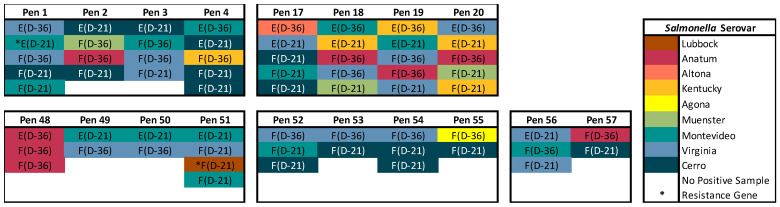
Fecal (F) and composite environment manure pack (E) isolates sequenced from Days -36 (D-36) and -21 (D-21) of the pre-trial period. The *Salmonella* serovars are indicated by color, and the presence of antimicrobial resistance genes are indicated by an asterisk. Pens 52 to 55 were selected for the manure slurry treatment.

**Figure 4 animals-13-03170-f004:**
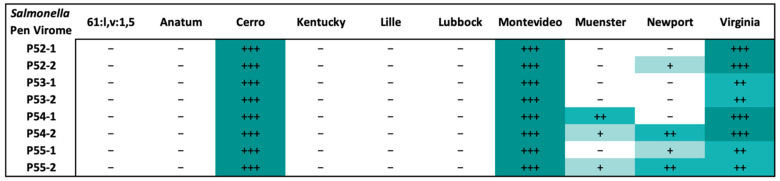
The *Salmonella* serovar host range of the virome from 4 WTAMU pens (P52 to P55), in duplicate (indicated by -1 or -2 after the pen number), determined by retrospective phage spot testing on the *Salmonella* serovars identified during the experimental trial. Results were assigned comparatively between plates ranging from the highest to lowest degrees of plaque clearing (+++, ++, +, -) which is indicated by the shading from darkest to lightest.

**Figure 5 animals-13-03170-f005:**
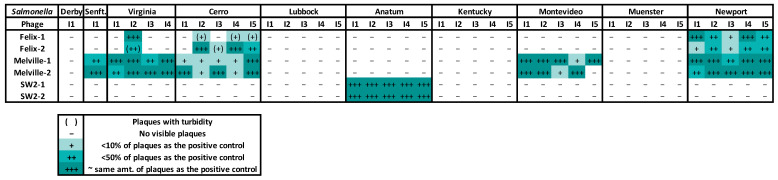
Results from routine test dilution (RTD) of selected phages on the 31 *Salmonella* isolates identified at the WTAMU and AgriLife feedlots. The darker shade of teal represents a higher degree (+++, ++, +) of bacterial cell clearing compared to no (-) clearing; results with turbidity (signifying partial host cell death) are signified with parentheses ( ).

**Figure 6 animals-13-03170-f006:**
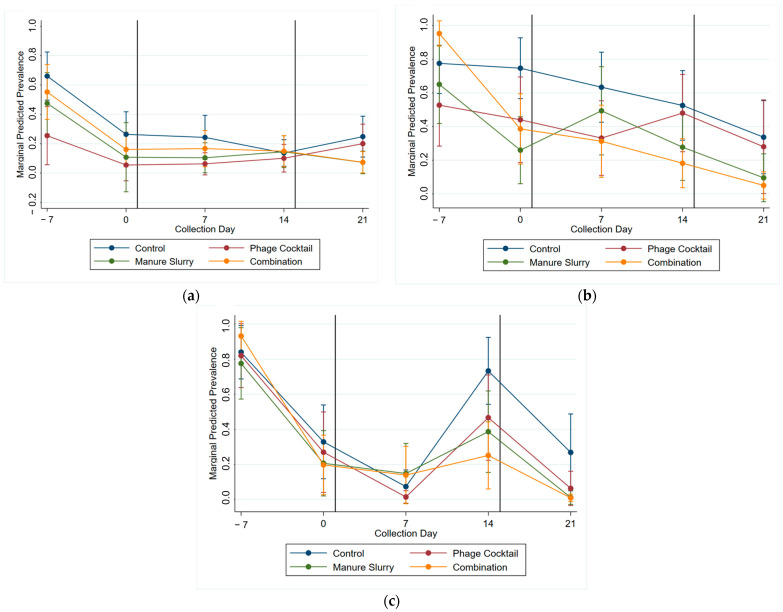
Marginal predicted *Salmonella* prevalence by treatment group (control—blue, manure slurry—green, phage cocktail—red, and combination—orange) and day for each cattle sample type: (**a**) feces, (**b**) brisket, and (**c**) rump. The vertical black line signifies the start of all treatment applications on Day 1 and the second phage cocktail treatment on Day 15.

**Figure 7 animals-13-03170-f007:**
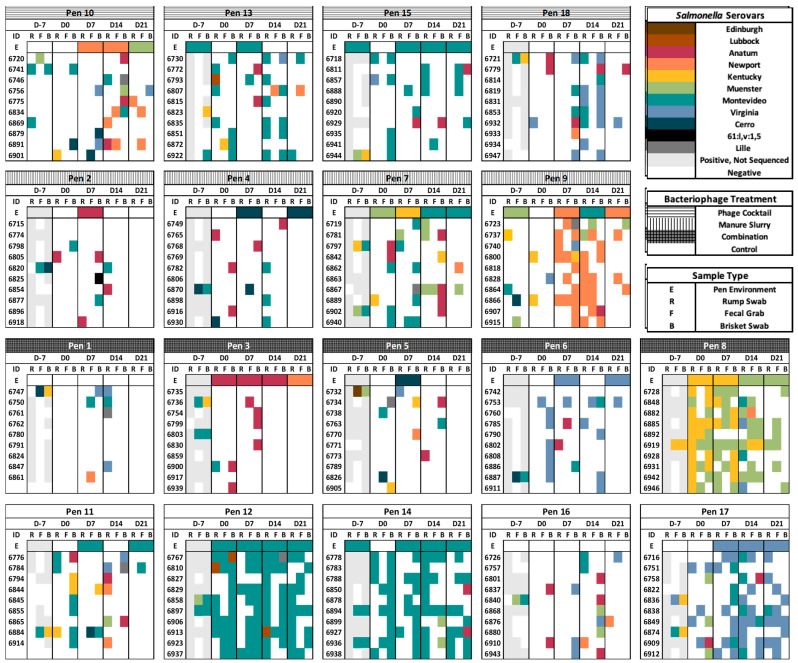
Samples (R—rump, F—feces, B—brisket, and E—composite environment manure pack) from Days -7 to 21 by feedlot pen, treatment, and cattle ID. Samples negative for the presence of *Salmonella* are white, samples that were positive for *Salmonella* but not sequenced are light gray, and the *Salmonella* serovars are signified by color. The pens in this graphic are arranged by treatment group, not according to the actual geographical location within the feedlot, but trends by feedlot pen can be observed in the phylogenetic tree. This figure’s design was adapted from a graphic by Levent et al. [20].

**Figure 8 animals-13-03170-f008:**
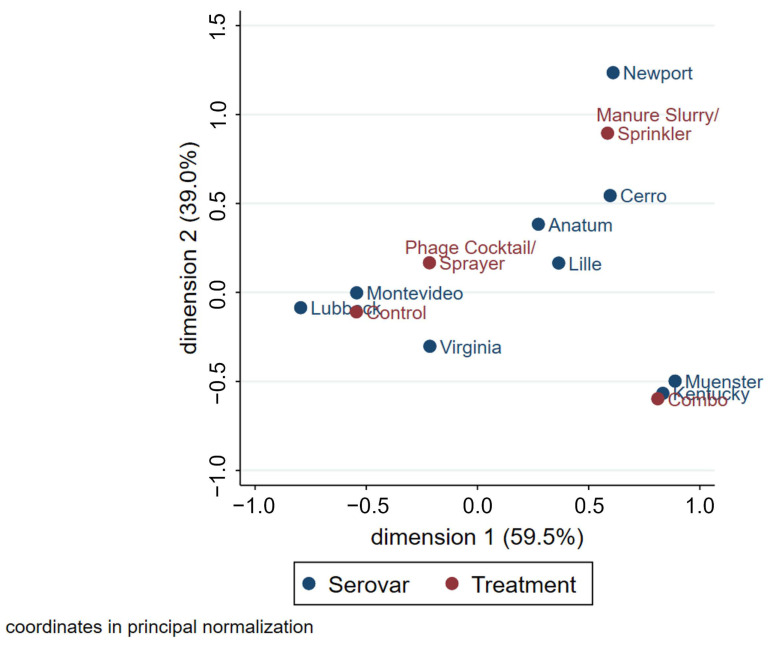
MCA plot of *Salmonella* serovars (blue) in relation to the treatment groups (red). Increased distance from the origin represents more distinct groups within the same variable. The angles from the origin between two different variables represent the strength of the relationship; the smaller the angle, the stronger the relationship.

**Figure 9 animals-13-03170-f009:**
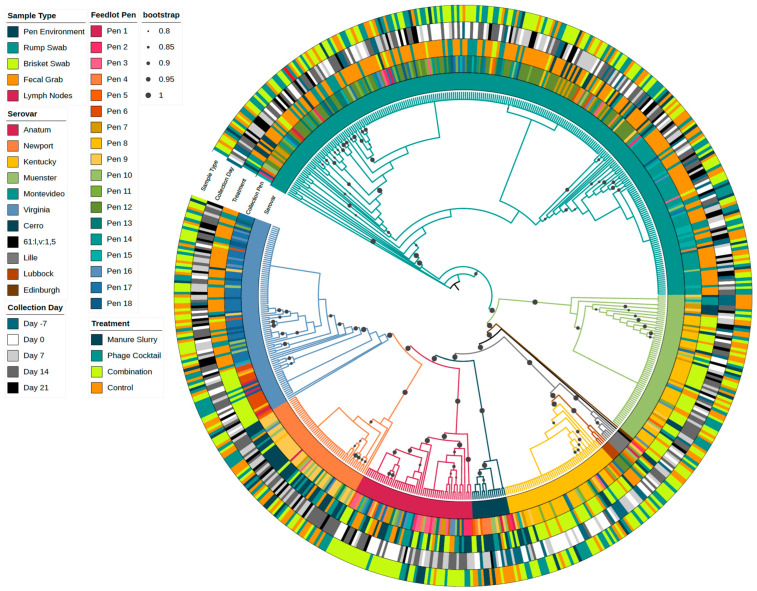
Phylogenetic tree of *Salmonella* isolates (n = 611). The inner ring represents the *Salmonella* serovars followed by the feedlot collection pen (shaded by block), treatment group, collection day (Day -7 of the pre-trial period and Days 0, 7, 14, and 21 of the study), and sample type. Bootstrap values ranging from 0.8 to 1 were included in the tree.

**Table 1 animals-13-03170-t001:** Bacteriophages used in the bacteriophage cocktail, the original stock titer, and final prepared titers for Day 1 and Day 15 applications during the experimental trial.

Bacteriophage	Phage Stock Titer (PFU/mL)	Day 1 Phage Final Titer (PFU/mL)	Day 15 Phage Final Titer (PFU/mL)
Melville	5.15 × 10^10^	1.02 × 10^9^	1.25 × 10^9^
Sw2	2.93 × 10^10^	7.70 × 10^8^	6.70 × 10^8^

**Table 2 animals-13-03170-t002:** *Salmonella* prevalence in cattle at the WTAMU and AgriLife feedlots during the follow-up period.

Month	Sample Type	WTAMU Feedlot PN02	AgriLife Feedlot
Aug	Feces	78.6% (22/28)	16.9% (30/178)
	Composite Environment	100.0% (3/3)	55.6% (10/18)
	Rump Swabs	N/A	2.2% (4/178)
Sept	Feces	85.7% (24/28)	7.9% (14/178)
	Composite Environment	100.0% (3/3)	27.8% (5/18)
	Rump Swabs	N/A	2.2% (4/178)
Oct	Feces	75.0% (21/28)	1.7% (3/178)
	Composite Environment	100.0% (3/3)	44.4% (8/18)
Nov	Feces	57.1% (16/28)	1.7% (3/178)
	Composite Environment	100.0% (3/3)	0.0% (0/0)
Dec	Feces	N/A ^a^	1.1% (2/178)
	Composite Environment ^b^	100.0% (3/3)	0.0% (0/0)
	Lymph Nodes	42.9% (6/14)	N/A
Jan	Feces	57.1% (8/14)	0.6% (1/177)
	Composite Environment ^b^	66.7% (2/3)	N/A
	Lymph Nodes	N/A	0.6% (1/176)
Feb	Feces	71.4% (10/14)	N/A
	Composite Environment ^b^	66.7% (2/3)	N/A
	Lymph Nodes	69.2% (9/13) ^c^	N/A

This includes prevalence data for all pens (n = 18) at the AgriLife Feedlot combined. N/A represent samples that were not collected; specific reasons besides the study design are included below. ^a^ Incorrect group of cattle sampled in December. ^b^ Cattle were moved from PNO2 to P14 to prepare for transport to slaughter; therefore, composite environment manure pack samples were collected from P14 not PNO2. ^c^ One carcass was railed off; therefore, the lymph node pair could not be collected.

## Data Availability

Sequencing data for this project can be found under NCBI BioProject accession number PRJNA807300.

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
