# Peer review of "An Experimental Field Trial Investigating the Use of Bacteriophage and Manure Slurry Applications in Beef Cattle Feedlot Pens for Salmonella Mitigation"

_animals, 2023, doi:10.3390/ani13203170_

Round 1
Reviewer 1 Report
This study explored the applications of manure slurry and bacteriophage cocktail treatments in beef cattle feedlot to reduced Salmonella.
Major concern: It was not sure what is the optimal time to give treatment.
Comments:Table 4, Four Salmonella serovars (Cerro, Virginia, Montevideo, and Agona) were identified in cattle feces from Pens 52 to 55, but the lysis of Salmonella serovar Agona by virome did not mentioned.
Reviewer 2 Report
Few suggestions-
Ln 127-131: When the author refers to isolates, is that phages or Salmonella isolates? Line 127 needs to be clarified. Give names for those five isolates that were selected for the study.
Ln 145: What are those cocktails? Made out of some Salmonella isolates (name?).
Ln 148: Why the soil extract was needed and must be clarified. Was it added before or after shaking?
Ln 267: Figure 2 and the paragraph is misleading. Pens receiving phage treatment shouldn't be shaded.
Table S7: Any explanation on why the combination group was not included in the multilevel mixed effects model? The reviewer sees Figure 6, but readers might be curious to examine the effect of the combined treatment in the pens.
Ln 484: Italicize Salmonella
The discussion sections should avoid portraying results (repeating) but instead, find relevant papers in the field and discuss what these results mean.
Reviewer 3 Report
This study, which investigates pre-harvest interventions for mitigating Salmonella in beef cattle, aligns well with the scope of the journal. The research explores the effectiveness of bacteriophages and manure slurry applications in feedlot pens to reduce Salmonella prevalence on cattle hides and potentially in lymph nodes. The findings suggest promising results for pre-harvest strategies aimed at enhancing food safety in beef production, making it highly relevant to the journal's objectives.
The hypothesis testability is well-established, and the methodology is generally sound, but some methodological inaccuracies and the absence of certain controls should be addressed for more robust results.
The introduction of this article could benefit from a more comprehensive literature review that contextualizes the research within the broader field of beef cattle farming and Salmonella mitigation. Additionally, providing a general introduction to the current status and practices in beef farming and management would enhance the reader's understanding of the study. To achieve this, consider referencing articles like " Beef Production in the Southwestern United States: Strategies Toward Sustainability " (https://doi.org/10.3389/fsufs.2020.00114) and “Health and welfare assessment of beef cattle during the adaptation period in a specialized commercial fattening unit” (https://doi.org/10.1016/j.rvsc.2023.03.008) for an overview of beef farming practices and challenges, I suggest citing both. This will help establish the relevance and importance of the research within the broader context of the beef industry.
Figure 1 is a crucial component of your paper, as it provides visual support for the concepts and data you are presenting. However, upon reviewing the figure, it appears to be of low quality, which can hinder readers' ability to fully comprehend the content.
Have you taken into account the dietary composition of the cattle during the trial, and have you explored the potential influences of this diet on the growth of Salmonella bacteria?
Have you conducted tests to determine the presence of mycotoxins in the provided diet?
Aflatoxins can have a multifaceted negative impact on beef cattle metabolism, including reduced feed intake, liver damage, impaired nutrient absorption, immunosuppression, decreased growth and performance, and the potential for mycotoxin residues in meat. Therefore, it is crucial to monitor and manage aflatoxin contamination in cattle feed to maintain animal health and optimize production.
I suggest the authors include a statement confirming that the levels of aflatoxin in their study were below the European Union (EU) limits, which are among the strictest in the world for mycotoxin contamination in animal feed. This will provide assurance regarding the safety of the feed used in their trial. I suggest citing this reference to support that (10.3390/toxins14070430).
Upon reviewing your study, it appears that there may be some issues with the clarity and alignment of your results with the materials and methods. It's essential for the results section to be a clear and logical extension of your study design. I suggest revisiting your results and ensuring that they are presented in a way that follows the sequence of your materials and methods. This will help readers better understand how your experimental design directly influenced the outcomes you observed. Clarifying this alignment will enhance the overall coherence of your study.
It's important to note that the discussion section of your paper lacks references to support the statements made. Throughout the discussion, you've presented several key points related to your results, but it would significantly enhance the quality and credibility of your paper if you could incorporate appropriate references to back up these statements. Providing citations for the points made will not only strengthen your arguments but also help readers connect your findings with existing literature. I recommend revisiting the discussion section and adding relevant references where necessary to provide a well-rounded context for your results.
Your study has provided valuable insights, but to make it more comprehensive, I recommend expanding the discussion on both the limitations and practical implications of your research. By addressing the study's limitations, you can provide a more balanced perspective on its scope and applicability. Additionally, elaborating on the practical implications will help readers understand how your findings can be translated into real-world scenarios or decision-making processes. This added depth will enhance the overall quality and usefulness of your work.
Round 2
Reviewer 3 Report
The authors have diligently addressed the review comments, significantly enhancing the paper's quality. As a result, it is now well-suited for publication.